Meta-analysis of northeast Atlantic marine taxa shows contrasting phylogeographic patterns following post-LGM expansions

Jenkins Tom L. 1
Castilho Rita 2
Stevens Jamie R. j.r.stevens@ex.ac.uk 1
1 Department of Biosciences, University of Exeter , Exeter , United Kingdom
2 Center for Marine Sciences, Campus de Gambelas, University of Algarve , Faro , Portugal
Costello Mark
Electronic publication date: 2018 Sep 28
Publication date: 2018
Volume: 6
Electronic Location ID: e5684
Received 2018 May 30; Accepted 2018 Aug 30
Copyright: ©2018 Jenkins et al.
Copyright year: 2018
Copyright holder: Jenkins et al.
License: This is an open access article distributed under the terms of the Creative Commons Attribution License, which permits unrestricted use, distribution, reproduction and adaptation in any medium and for any purpose provided that it is properly attributed. For attribution, the original author(s), title, publication source (PeerJ) and either DOI or URL of the article must be cited.
License URL: https://creativecommons.org/licenses/by/4.0/

Keywords: Comparative phylogeography, Historical demography, Last Glacial Maximum, mtDNA, Northeast Atlantic, Population expansion

Funding: Natural Environment Research Council (UK) NE/L002434/1 Natural England PO 904130 FCT—Foundation for Science and Technology UID/Multi/04326/2013 This research was funded by a Natural Environment Research Council (UK) GW4+ DTP studentship (Grant No. NE/L002434/1), Natural England (Ref. PO 904130) and the University of Exeter. Rita Castilho was funded by FCT—Foundation for Science and Technology through project UID/Multi/04326/2013. The funders had no role in study design, data collection and analysis, decision to publish, or preparation of the manuscript.

==============================
Background

Comparative phylogeography enables the study of historical and evolutionary processes that have contributed to shaping patterns of contemporary genetic diversity across co-distributed species. In this study, we explored genetic structure and historical demography in a range of coastal marine species across the northeast Atlantic to assess whether there are commonalities in phylogeographic patterns across taxa and to evaluate whether the timings of population expansions were linked to the Last Glacial Maximum (LGM).

Methods

A literature search was conducted using Web of Science. Search terms were chosen to maximise the inclusion of articles reporting on population structure and phylogeography from the northeast Atlantic; titles and abstracts were screened to identify suitable articles within the scope of this study. Given the proven utility of mtDNA in comparative phylogeography and the availability of these data in the public domain, a meta-analysis was conducted using published mtDNA gene sequences. A standardised methodology was implemented to ensure that the genealogy and demographic history of all mtDNA datasets were reanalysed in a consistent and directly comparable manner.

Results

Mitochondrial DNA datasets were built for 21 species. The meta-analysis revealed significant population differentiation in 16 species and four main types of haplotype network were found, with haplotypes in some species unique to specific geographical locations. A signal of rapid expansion was detected in 16 species, whereas five species showed evidence of a stable population size. Corrected mutation rates indicated that the majority of expansions were estimated to have occurred after the earliest estimate for the LGM (∼26.5 Kyr), while few expansions were estimated to have pre-dated the LGM.

Conclusion

This study suggests that post-LGM expansion appeared to be common in a range of marine taxa, supporting the concept of rapid expansions after the LGM as the ice sheets started to retreat. However, despite the commonality of expansion patterns in many of these taxa, phylogeographic patterns appear to differ in the species included in this study. This suggests that species-specific evolutionary processes, as well as historical events, have likely influenced the distribution of genetic diversity of marine taxa in the northeast Atlantic.

Introduction

Comparative phylogeographic studies present opportunities to explore how historical events may have helped shape patterns of genetic structure amongst co-distributed species (Avise et al., 1987; Avise, 2009; Hickerson et al., 2010). Patterns of concordant phylogeographical structure across multiple taxa are particularly informative because, while some patterns of spatial genetic structure may be caused by species-specific evolutionary processes, patterns common across multiple taxa may suggest similar evolutionary histories, such as common barriers to gene flow (Avise, 2009; Hickerson et al., 2010). These findings can be important for conservation because of the potential to modify management actions in the light of the differing phylogeography of multiple species across the same geographical area (Pelc, Warner & Gaines, 2009; Toonen et al., 2011; Heyden et al., 2014; Liggins et al., 2016). In marine biology, such comparative studies have made important contributions to our understanding of how historical events, such as the Pleistocene glaciations, have helped shape the spatial patterns of contemporary genetic diversity of marine taxa (Patarnello, Volckaert & Castilho, 2007; Maggs et al., 2008; Marko et al., 2010; Ni et al., 2014).

The Pleistocene epoch was characterised by recurrent glaciations and intensive fluctuations in climate that periodically influenced the spatial distributions of plants and animals (Hewitt, 1999; Hofreiter & Stewart, 2009). The most recent glacial period began approximately 115 Ka and nearly all ice sheets were at their maximum (Last Glacial Maximum, LGM) between 26.5–19 Ka (Clark et al., 2009). The advances of the Northern Hemisphere ice sheets led to significant changes in temperature and sea levels (Lambeck & Chappell, 2001). This must have had profound implications for habitat availability and the population persistence of coastal species—large parts of species’ ranges would have been reduced, while other species may have survived in glacial refugia (Maggs et al., 2008; Provan & Bennett, 2008). As the ice retreated and the sea level rose, a number of individuals from refugial populations may have dispersed and recolonised areas unavailable during the glaciation (Hewitt, 2000). Changes in latitudinal ranges and population sizes can have distinct effects on the genetic architecture of a species due to the competing processes of mutation, drift and selection; moreover, the deep molecular divergence reported in taxa associated with several known European refugia suggests repeated expansion and contraction of conspecific populations were common throughout the Pleistocene (Hewitt, 2004).

In the northeast Atlantic, the ice sheets extended as far south as Britain and Ireland, leaving an ice-free zone in mid-southern England, with possibly a small area in southwest Ireland free of ice (Chiverrell & Thomas, 2010). However, the predicted extent of ice coverage across southern Ireland and the Celtic Sea differs among studies (e.g., Taberlet et al., 1998; Hughes et al., 2016). The advance of the ice sheets led to a drastic drop in sea levels in the English Channel, resulting in the complete emersion of the channel between England and France, except for a palaeo-river that extended across the continental margin (Ménot et al., 2006). This suggests that extant coastal communities inhabiting these areas are likely recolonisers originating from glacial refugia. It has been suggested that Hurd Deep, a trench in the English Channel (Fig. 1), might have persisted as a marine lake during the LGM, thereby acting as a potential glacial refugium (Provan, Wattier & Maggs, 2005; Hoarau et al., 2007). Other areas further south, including Brittany (Coyer et al., 2003) and the Iberian Peninsula (Hoarau et al., 2007; Neiva et al., 2012) (Fig. 1), have also been postulated to act as refugia during the LGM. This was supported by high levels of genetic diversity found at these areas in the species studied, a key signature indicative of glacial refugia (Provan & Bennett, 2008).

Figure 1 Topographical map of the northeast Atlantic Ocean.

The white dotted lines represent the maximum extent of ice cover during the Last Glacial Maximum (LGM) (redrawn from Hughes et al., 2016). Orange lines indicate putative refugia: Hurd Deep, Brittany and Iberia.

Studies of single-species phylogeography across the northeast Atlantic are common; yet, because of the differences in molecular methodologies and analytical approaches, it can be difficult to compare results reliably. By applying a consistent methodology across all studies, this standardises the analysis (Harrison, 2011), enabling patterns of phylogeography to be explored and compared within and across taxa. Two comparative meta-analyses in the Atlantic Ocean have been published to-date: the first explored the feasibility of distinguishing genetic signatures of periglacial refugia from southern refugia in eight benthic marine species (Maggs et al., 2008), and the second looked for concordance among phylogeographical breaks around the southeast coast of the United States of America (Pelc, Warner & Gaines, 2009). Systematic meta-analyses across diverse taxa in other seas and oceans have proved useful for exploring broad patterns of phylogeography (e.g., Patarnello, Volckaert & Castilho, 2007; Kelly & Palumbi, 2010; Marko et al., 2010; Ni et al., 2014); for example, one study of rocky-shore taxa from the northeastern Pacific found that 36% of species showed evidence of population expansions associated with the LGM, while 50% exhibited demographic patterns consistent with stable effective population sizes (Marko et al., 2010). However, such a study for marine taxa across the northeast Atlantic has yet to be undertaken.

In this study, we reanalyse available mitochondrial (mt)DNA data to compare the phylogeography of coastal benthic and demersal organisms across the northeast Atlantic (Fig. 1), an area characterised by complex oceanography and historical biogeographical events, such as the Pleistocene glaciations. Specifically, our aims were: (i) to identify commonalties (or otherwise) in contemporary genetic structure; (ii) to re-examine historical demography to test for signatures of population expansions; and (iii) to estimate the timings of any expansions detected. We discuss our findings in the context of the Pleistocene glaciations, asking in particular whether the LGM affected the phylogeography of marine taxa concordantly or discordantly.

Material and Methods

Literature search

To compare the phylogeography of benthic and demersal organisms across the northeast Atlantic, we undertook a meta-analysis of molecular phylogeographic studies. A literature search was conducted by TLJ and JRS using Web of Science (Thomson Reuters) in February 2015. Search terms were chosen to maximise the inclusion of articles reporting on population structure and phylogeography from the northeast Atlantic. The following sets of Boolean search terms were submitted to the Advanced Search Tool: (1) gene flow OR population structure OR genetic diversity OR phylogeograph*; (2) marine OR intertidal OR subtidal OR estuar*; and (3) Atlantic. Titles and abstracts were screened by TLJ and JRS to identify suitable articles within the scope of this study and only articles that matched the following criteria were retained: (a) organisms were fully marine or estuarine throughout their life history (diadromous species were excluded); (b) studies of temporal changes, hybridisation or introgression from closely related species were omitted; (c) the study included at least three sampling sites from within the northeast Atlantic (Fig. 1—sites outside of this area were not considered); (d) datasets contained a minimum of five individuals per site and a total sample size of at least 50; and (e) the study included latitude and longitude of the sampling sites or a detailed description or map which provided sufficient detail to determine the geographical location of sample origins. The studies were reviewed independently by TLJ and JRS and there were no disputes regarding inclusion or rejection that needed adjudication. Given the proven utility of mtDNA in comparative phylogeography (e.g., Patarnello, Volckaert & Castilho, 2007; Ni et al., 2014) and the availability of these data in the public domain, a meta-analysis was conducted using published mtDNA gene sequences.

Data reanalysis

A standardised methodology was implemented to ensure that all mtDNA datasets were reanalysed in a consistent and directly comparable manner. Data analyses in the original studies were far from consistent, particularly with respect to the analysis of haplotype networks and historical demography. The majority of studies reported information about population structure, however, in several instances the studies included additional samples outside of the northeast Atlantic in their analysis. Therefore, standardised tests of population structure were undertaken de novo for each species. Sites that were genetically homogeneous (as described by the original authors) and which were spatially close or situated in the same geographical region were combined in some datasets. This ensured that phylogeography within and across seas was examined in this meta-analysis. Population differentiation was examined using global values of Jost’s D (Jost, 2008) and FST (Weir & Cockerham, 1984) using the fastDivPart function from the R package diveRsity (Keenan et al., 2013; R Core Team, 2016) and significance was assessed using 10,000 permutation replicates.

To examine the genealogical relationships within species, haplotype networks were constructed using the haploNet function from the R package pegas (Paradis, 2010). Tajima’s D (Tajima, 1989), Fu’s FS (Fu, 1997) and Ramos-Onsins’ R 2 (Ramos-Onsins & Rozas, 2002) neutrality tests were performed in DnaSP v5.10 (Librado & Rozas, 2009) to determine whether each species carried a signal that deviated from neutrality (significance was assessed using 10,000 bootstrap replicates). Mismatch analyses (frequency of pairwise nucleotide-site differences between sequences) were carried out using the population growth-decline model in DnaSP to further examine the demographic history, and Harpending’s raggedness index (r) (Harpending, 1994) was used to evaluate the fit of the observed distribution to the growth-decline model (10,000 bootstrap replicates). A non-significant index suggests that the observed data have a relatively good fit to the growth-decline model. In contrast, a significant index is indicative of a stable population which is typically thought to show a ‘ragged’, multi-modal mismatch (Harpending, 1994).

The equation t = τ/(2µk) was used to estimate the timing of a population expansion (t), where τ is the date of the expansion measured in units of mutational time (Tau –estimated using DnaSP), µis the mutation rate per site per year and k is the sequence length. In addition, Bayesian Skyline Plots (BSPs) were run using BEAST2 v2.5.0 (Drummond et al., 2005; Bouckaert et al., 2014). BEAST2 uses a Markov chain Monte Carlo (MCMC) sampling procedure to estimate effective population size (Ne) through time based on the temporal distribution of coalescences in gene genealogies. For each dataset, the substitution model was selected using bModelTest (Barido-Sottani et al., 2018), which uses reversible jump MCMC that allows the Markov chain to jump between states representing different possible substitution models. A strict clock and a coalescent Bayesian Skyline prior was implemented. Each run consisted of 100 million steps with a burn-in of one million and parameters were sampled every 10,000 steps. Chain convergence and BSPs were analysed with Tracer v1.7.1 (Rambaut et al., 2018).

Recent studies have shown that the use of mutation rates derived from ancient calibration dates or from phylogenetic analyses may not be appropriate for studies at the population level (Ho et al., 2008; Ho et al., 2011). In this study, therefore, mutation rates were chosen based on the most recent calibration date available for the closest taxonomic relative (Table S1). In published studies where a mutation rate was not specified, the genetic distance provided by the study was divided by the date of the calibration event (in Myr) to obtain a % mutation rate per Myr. For cases where only calibration dates older than 5 Myr were available for the species and gene of interest, a three-fold correction in mutation rate was applied to the original rate to control for the potential time-dependency of molecular rates. This adjustment was implemented because rates have been found to vary by three to six-fold for several marine species when calibration dates younger than 5 Myr vs. older dates have been tested (Crandall et al., 2012; Laakkonen, Strelkov & Väinölä, 2015). A range of mutation rates based on the rates reported by previous studies were used to calculate a minimum, maximum and average time estimate since a population expansion.

Results

Literature search

The initial search using Boolean terms identified 1,120 articles, which was reduced to 56 articles after the titles and abstracts were examined and the search criteria were applied (Fig.  S1). The final database for the meta-analysis consisted of mtDNA gene sequence data from 21 studies (Table 1); some studies from the previous step were not included due to the use of RFLPs in mtDNA or because some mtDNA datasets were not publicly available. The final database spanned several taxonomic groups, with fishes, molluscs and crustaceans accounting for the majority of species (81%). The most common mitochondrial gene across all studies was cytochrome oxidase I (COI), followed by cytochrome b (Cyt b), the control region (CR) and the intergenic spacer region (IGS). COI was the most commonly used gene for invertebrate studies, IGS for macroalgae, and studies of fish used either the CR or the Cyt b gene.

Table 1 List of the papers used in the meta-analysis and a summary of the information extracted from each study.

Taxonspecies	MtDNAgene	No. sites;N	Sampling site distribution	Larval development	No. of lineages	Reference	
Crustacean							
Carcinus maenas	COI	13; 200	SW Spain to Norway	PLD, long	1	Roman & Palumbi (2004)	
Maja brachydactyla	COI	13; 291	SW Spain to W Ireland	PLD, 2–3 wk	1	Sotelo et al. (2008)	
Neomysis integer	COI	9; 379	SW Spain to E Scotland	No PLD, brooder	1	Remerie et al. (2009)	
Palinurus elephas	COI	6; 119	S Portugal to W Scotland	PLD, up to 1 yr	1	Palero et al. (2008)	
Fish							
Conger conger	CR	4; 232	Azores to Ireland	Leptocephalus, up to 2 yr	1	Correia et al. (2012)	
Dicentrarchus labrax	CR	9; 93	Bay of Biscay to Norway	PLD, 8–12 wk	1	Coscia & Mariani (2011)	
Labrus bergylta	CR	7; 279	W Ireland to Norway	PLD, 37–49 d	1	D’Arcy, Mirimin & FitzGerald (2013)	
Pomatoschistus microps	Cyt b	10; 232	Bay of Biscay to Norway	PLD, 6–9 wk	1	Gysels et al. (2004)	
Pomatoschistus minutus	Cyt b	8; 165	S Portugal to Norway	PLD, unknown	1	Larmuseau et al. (2009)	
Raja clavata	Cyt b	9; 315	Azores to North Sea	No PLD, oviparous	1	Chevolot et al. (2006)	
Solea solea	Cyt b	10; 645	Bay of Biscay to Skagerrak	PLD, up to 3 wk	1	Cuveliers et al. (2012)	
Symphodus melops	CR	10; 263	S Portugal to Skagerrak	PLD, 14–25 d	1	Robalo et al. (2012)	
Macroalgae							
Pelvetia canaliculata	IGS	15; 429	Portugal to Norway	No PLD, external fertilisation	1	Neiva et al. (2014)	
Mollusc							
Cerastoderma edule	COI	12; 300	Portugal to Norway	PLD, up to 4 wk	1	Krakau et al. (2012)	
Macoma balthica	COI	15; 339	Bay of Biscay to North Sea	PLD, 2–5 wk	2	Becquet et al. (2012)	
Modiolus modiolus	COI	4; 73	Irish Sea to Norway	PLD, up to 24 wk	2	Halanych et al. (2013)	
Nassarius nitidus	COI	3; 62	NW Spain to Sweden	PLD, 4–8 wk	1	Couceiro et al. (2012)	
Nassarius reticulatus	COI	6; 156	S Portugal to UK	PLD, 4–8 wk	1	Couceiro et al. (2007)	
Polychaete							
Owenia fusiformis	COI	11; 283	Portugal to North Sea	PLD, up to 28 d	3	Jolly et al. (2005)	
Pectinaria koreni	COI	10; 289	Portugal to North Sea	PLD, up to 15 d	2	Jolly et al. (2006)	
Bryozoan							
Celleporella hyalina	COI	9; 63	NW Spain to Iceland	PLD, 1–4 h	1	Gómez et al. (2007)	
Notes.

MtDNA mitochondrial DNA

No. of sites number of sampling sites

N total number of sequenced individuals

PLD pelagic larval duration

Genetic structure

Sixteen species showed significant global Jost’s D and FST values, indicative of population differentiation (Table 2), while the remaining five species showed little evidence of population differentiation. Across the 21 datasets, four different types of haplotype network were putatively identified based on the structure of the networks (Fig. 2) (all haplotype networks are presented in Fig. S2):

Table 2 Summary statistics for each species. Population differentiation and demographic statistics are shown. In all statistical tests, significance was assessed using 10,000 permutations or bootstraps replicates.

Species	Population differentiation		Demography	
	Jost’sD	FST		Tajima’sD	FS	R2	r	Expansion	
Crustacean									
Carcinus maenas	0.584***	0.157***		−1.73*	−40.36***	0.034*	0.018	Yes	
Maja brachydactyla	0.298***	0.045***		−1.86**	−33.72***	0.028*	0.030	Yes	
Neomysis integer	0.956***	0.554***		0.14	−0.954	0.024	0.086	No	
Palinurus elephas	0.023	0.000		−2.31***	−30.19***	0.019*	0.094	Yes	
Fish									
Conger conger	0.124	0.000		−2.58***	−211.1***	0.012***	0.031	Yes	
Dicentrarchus labrax	0.540*	0.031*		−1.88**	−21.52***	0.047*	0.011	Yes	
Labrus bergylta	0.672***	0.135***		−0.53	−49.35***	0.074	0.024	Yes	
Pomatoschistus microps	0.391***	0.385***		−1.39	−17.90***	0.044	0.215	Yes	
Pomatoschistus minutus	0.652***	0.100***		−1.96**	−90.56***	0.034*	0.015	Yes	
Raja clavata	0.375***	0.330***		−0.09	−2.340	0.076	0.309	No	
Solea solea	0.049	0.002		−2.02***	−131.9***	0.021**	0.221	Yes	
Symphodus melops	0.578***	0.349***		−1.70*	−50.52***	0.032*	0.086	Yes	
Macroalgae									
Pelvetia canaliculata	0.689***	0.482***		−1.53*	−19.02***	0.036	0.043	Yes	
Mollusc									
Cerastoderma edule	0.662***	0.304***		−2.24***	−34.47***	0.019**	0.033	Yes	
Macoma balthica	0.702***	0.470***		–	–	–	–	–	
lineage 1	–	–		−0.80	−3.773	0.053	0.241	No	
lineage 2	–	–		−0.99	−1.110	0.089	0.173	No	
Modiolus modiolusa	0.083	<0.001		−1.79*	−11.91***	0.045*	0.156	Yes	
Nassarius nitidus	0.222***	0.302***		−1.49*	0.028	0.049*	0.446	No	
Nassarius reticulatus	0.047	0.000		−2.51***	−48.33***	0.016**	0.080	Yes	
Polychaete									
Owenia fusiformis	0.788***	0.055***		–	–	–	–	–	
lineage 1	–	–		−2.34***	−114.8***	0.024**	0.020	Yes	
lineage 2	–	–		−2.06**	−55.00***	0.030**	0.008**	Yes	
lineage 3	–	–		−1.26	−3.934**	0.084	0.080	Yes	
Pectinaria koreni	0.596***	0.112***		–	–	–	–	–	
lineage 1	–	–		−1.99**	−76.48***	0.027**	0.021	Yes	
lineage 2	–	–		−2.63***	−54.02***	0.018***	0.029*	Yes	
Bryozoan									
Celleporella hyalina	0.513***	0.488***		−1.35	−0.554	0.063	0.061	No	
Notes.

* <0.05.

** <0.01.

*** <0.001.

Fs Fu’s Fs

R2 Ramos-Onsins’ R 2

r Harpending’s raggedness index

a Only statistics for lineage 1 are shown.

Figure 2 Haplotype networks showing four different network structures.

Haplotype networks showing (A) ‘star’ (Palinurus elephas), (B) ‘complex star’ (Carcinus maenas), (C) ‘reciprocally monophyletic’ (Macoma balthica) and (D) ‘complex mutational’ (Dicentrarchus labrax) structures. Each circle represents a unique haplotype and the sizes of the circles are proportional to the haplotype frequencies for each network but are not comparable across studies. Each line represents one mutation step and two or more steps are indicated by bars or numbers. Colours inside the circles correspond to sites which have individuals represented in that particular haplotype. Species illustrations by Guy Freeman.

(i) A ‘Star’ network (Fig. 2A), in which a single, widespread haplotype is typically positioned at the centre of the network and is thought to be the ancestral haplotype. Additional haplotypes are linked to this dominant haplotype by a single (or a few) mutational step(s), suggesting these haplotypes are the product of recent mutation events. Eight species showed this type of relationship (Celleporella hyalina, Conger conger, Nassarius nitidus, Nassarius reticulatus, Palinurus elephas, Pelvetia canaliculata, Pomatoschistus microps and Raja clavata). In one case, the dominant haplotype had far fewer connections than a low-frequency haplotype in the network, making it difficult to distinguish the centre of the network with confidence (Pomatoschistus microps);

(ii) A ‘Complex star’ network (Fig. 2B), in which there are multiple high-frequency haplotypes and connections. Six species showed this type of relationship (Carcinus maenus, Cerastoderma edule, Maja brachydactyla, Pomatoschistus minutus, Solea solea, Symphodus melops);

(iii) A ‘Reciprocally monophyletic’ network (Fig. 2C), in which more than one lineage is apparent and each lineage is linked by a long branch associated with numerous mutations. Four species showed this type of relationship (Macoma balthica, Modiolus modiolus, Owenia fusiformis and Pectinaria koreni);

(iv) A ‘Complex mutational’ network (Fig. 2D), in which some branches were separated by a very large number of mutations, while other branches had contrarily one or two mutations. Three species showed this type of relationship (Dicentrarchus labrax, Labrus bergylta and Neomysis integer). In most cases, a dominant haplotype was present and was presumed to be the ancestral form. However, Neomysis integer presented an unusual network in which a distinct ancestral haplotype was not apparent and the centre of the haplotype network was not readily distinguishable.

Historical demography

Historical demography was inferred for each species based on the observed mismatch distribution, neutrality tests and the raggedness index (Table 2). Four main types of mismatch distributions were observed: unimodal, skewed unimodal, multimodal and bimodal (Fig. 3) (all mismatch distributions are presented in Fig. S3). Unimodal is associated with a sudden population expansion (e.g., Maja brachydactyla; Fig. 3A), and skewed unimodal is generally associated with a recent expansion or bottleneck (e.g., Nassarius reticulatus; Fig. 3B). Multimodal (e.g., Labrus bergylta; Fig. 3C) and bimodal (e.g., Macoma balthica; Fig. 3D) are usually associated with constant population size. However, previous research has suggested that bimodal peaks may indicate the presence of two distinct lineages (e.g., Alvarado-Bremer et al., 2005), which would potentially violate the assumptions of coalescent theory if analysed as one ‘genetic’ population. In this case, the first peak would represent intra-clade pairwise differences, whereas the second peak would likely represent more ancient inter-clade pairwise differences (Fig. 3D). For each instance of bimodality, the haplotype network was inspected for evidence of two or more lineages. The networks indicated that more than one distinct lineage was evident for all bimodal mismatches (Macoma balthica, Modiolus modiolus, Owenia fusiformis and Pectinaria koreni) and, therefore, mismatch analysis and neutrality tests were carried out on each lineage separately. These analyses were not conducted for lineage 2 of Modiolus modiolus due to the small number of individuals (N = 3) comprising this lineage.

Figure 3 Mismatch distributions showing four different distributions.

Mismatch distributions showing (A) unimodal (Maja brachydactyla), (B) skewed unimodal (Nassarius reticulatus), (C) multimodal (Labrus bergylta) and (D) bimodal (Macoma balthica). Unimodal and skewed unimodal distributions are generally associated with a sudden expansion and a recent sudden expansion, respectively. Multimodal and bimodal are thought to be associated with a constant population size (but see text). Bars represent the frequency of pairwise nucleotide differences between individuals. Curves correspond to the expected distribution fitted to the data under a model of constant population size (solid line) or demographic expansion (dotted line). Species illustrations by Guy Freeman.

Neutrality statistics for testing the drift–mutation equilibrium (Tajima’s D, FS and R 2) were found to be contrasting between species (Table 2). These tests tended to be significant for species that showed a star-shaped network and for which the mismatch graph was unimodal or skewed unimodal. This supported evidence that a signal of rapid population expansion was detected; however, a selective sweep can also produce the same genetic signal. Harpending’s r suggested that two datasets departed from a model of demographic expansion (Table 2), but inspection of the mismatch graphs and neutrality tests indicated there was strong evidence to support a rapid population expansion (or selective sweep) in both datasets. No signatures of rapid population expansion were detected in five species (Celleporella hyalina, Macoma balthica, Nassarius nitidus, Neomysis integer and Raja clavata), suggesting a stable constant population size.

For the remaining 19 datasets (16 species, 19 including lineages), a historic population expansion was assumed and the timing of the expansion was estimated (Fig. 4). All expansions were found to take place during the Pleistocene or the Holocene epoch. Estimated timings for 17 datasets were after or overlapped the earliest estimate for the LGM (∼26.5 Ka). Expansion estimates for one fish (Labrus bergylta) and one lineage of the polychaete Owenia fusiformis pre-dated the LGM but were still positioned during the last glacial period. Bayesian Skyline Plots (Fig. 5) were generally consistent with the results from the mismatch analyses. Among the 17 datasets for which from the mismatch analyses expansion times were estimated to have occurred after the LGM, a rise in Ne post-LGM was apparent in 15 of these datasets, but the strength of the increase varied across datasets. In comparison to the mismatch analysis, the BSP for L. bergylta (Fig. 5F) and O. fusiformis lineage 2 (Fig. 5P) indicated a population expansion after the earliest estimate for the LGM as opposed to pre-dating the LGM. In addition, although the mismatch analyses inferred a post-LGM expansion for M. modiolus lineage 1 (Fig. 5M) and O. fusiformis lineage 3 (Fig. 5Q), BSPs generally suggested Ne was constant after the LGM.

Figure 4 Estimated dates of expansion for species or lineages (L) in which the demographic expansion hypothesis was not rejected.

A minimum and maximum time since expansion is plotted as horizontal bars for some datasets, estimated from a minimum and maximum mutation rate (Table S1). The beginning of the last glacial period (dotted line) and the estimated time-frame of the Last Glacial Maximum (grey shaded area) are displayed. Species are organised by taxa: crustaceans, Carcinus maenas –Palinurus elephas); fish, Conger conger –Symphodus melops; macroalgae, Pelvetia canaliculata; molluscs, Cerastoderma edule –Nassarius reticulatus; polychaetes, Owenia fusiformis –Pectinaria koreni. Species illustrations by Guy Freeman.

Figure 5 Bayesian Skyline Plots for species or lineages (L) in which the demographic expansion hypothesis was not rejected.

Solid black lines show the medium effective population size over time (Ne = effective population size and T = generation time); dashed black lines represent the 95% confidence intervals. The estimated time-frame of the Last Glacial Maximum is denoted by the area shaded dark grey. Species are organised by taxa: crustaceans, Carcinus maenas (A), Maja brachydactyla (B), Palinurus elephas (C); fish, Conger conger (D), Dicentrarchus labrax (E), Labrus bergylta (F), Pomatoschistus microps (G), P. minutus (H), Solea solea (I), Symphodus melops (J); macroalgae, Pelvetia canaliculata (K); molluscs, Cerastoderma edule (L), Modiolus modiolus Lineage 1 (M), Nassarius reticulatus (N); polychaetes, Owenia fusiformis Lineage 1 (O), O. fusiformis Lineage 2 (P), O. fusiformis Lineage 3 (Q), Pectinaria koreni Lineage 1 (R), P. koreni Lineage 2 (S).

Discussion

The results of this study show a range of contemporary genetic patterns across the coastal marine taxa analysed in the northeast Atlantic. In general, genealogical patterns were not uniform within taxonomic groups, though common patterns were observed in both polychaete species, which implies that historical events may have affected these polychaete species similarly. Most species (76%) showed evidence of population structuring, suggestive of restricted contemporary or historical gene flow between the sites studied. Of the species that exhibited no population differentiation, all five species have a pelagic larval phase, with a pelagic larval duration (PLD) ranging from up to three weeks (S. solea) to a year or more (P. elephas and C. conger) (Table 1). However, most of the species that demonstrated significant population differentiation also had a pelagic larval phase, ranging from a relatively short PLD of 1–4 h (C. hyalina) to a relatively long PLD of 8–12 weeks (D. labrax) (Table 1). Although speculative, taken altogether, this may suggest that larval development and PLD could be important factors in maintaining gene flow in some, but not all, of these species; however, more evidence is needed to confirm this. Indeed, whether a general correlation exists between PLD and genetic differentiation measures remains unclear because some studies have reported poor correlations between the two (Weersing & Toonen, 2009; Kelly & Palumbi, 2010; Riginos et al., 2011), while other studies have reported the opposite (Siegel et al., 2003; Selkoe & Toonen, 2011) suggesting that PLD and genetic metrics can indeed reflect scales of dispersal if the sampling design is robust (Selkoe & Toonen, 2011). As a result, speculative relationships between PLD and genetic differentiation should be interpreted with caution.

In some of the species studied, certain geographical areas were dominated by a particular haplotype that was rarely or not present in other areas across the sampled range. For example, the green crab Carcinus maenas showed highly significant differentiation and distinctive haplotypes in the Faroe Islands and Iceland, a pattern detected by the original authors who subsequently concluded that a deep-water barrier to dispersal in green crabs was the driver of this pattern (Roman & Palumbi, 2004). A similar pattern was also observed for two species around western Ireland in the northeast Atlantic. In Celleporella hyalina and Macoma balthica, distinct haplotypes composed a population around western Ireland; however, unique haplotypes were not apparent in other species analysed in this study with similar sampling coverage (e.g., Labrus bergylta, Palinurus elephas and Pelvetia canaliculata). A discrepancy in genetic structure between species at this spatial scale has also been observed between two temperate octocoral species (Eunicella verrucosa and Alcyonium digitatum) using microsatellite markers, whereby northwest Ireland samples were found to be genetically isolated from other northeast Atlantic samples in E. verrucosa, but not in A. digitatum (Holland, Jenkins & Stevens, 2017). This suggests that historical or contemporary gene flow between areas in the northeast Atlantic and western Ireland is likely possible, but in some cases the spatial patterns of genetic structure could be influenced by other processes such as strong selection pressures, species-specific life history traits, demographic fluctuations, or range expansions occurring at different times in different species (Hellberg, 2009).

Demographic history

Demographic history was variable across species in the northeast Atlantic, as evidenced by both the diverse structuring of the haplotype networks and the observed mismatch distributions within species. The presence of one or more lineages and the complexity of mutational patterns in several networks suggested some species have undergone pronounced changes in their demography and genealogy. Connections with large mutation steps separating some haplotypes are indicative of deep phylogenetic splits in the genealogies and suggests the persistence of old populations in these species. Accumulating new mutations is a relatively slow process and, therefore, sufficient time since coalescence must have elapsed to facilitate these large sequence divergences (Avise, 2009).

In the northeast Atlantic, the LGM has often been viewed as a possible explanation for discrepancies in genealogies and for rapid population expansions via recolonisation as glaciers started to retreat from their maximum positions (Hewitt, 2004). In this study, we detected rapid expansions in many different taxa, of which the majority were estimated to occur after the LGM. This supports evidence for post-LGM expansions, possibly from periglacial refugia (Maggs et al., 2008) or via recolonisation of areas previously affected by the Northern Hemisphere ice sheets. These results are in contrast to the northeast Pacific where regional persistence during the LGM appeared to be common in rocky-shore organisms (Marko et al., 2010). The conclusions of several previous studies reanalysed in this meta-analysis also detected rapid expansions (e.g., Jolly et al., 2006; Sotelo et al., 2008; Larmuseau et al., 2009); however, the authors of these studies estimated the dates of these expansions to have occurred pre-LGM. This discrepancy could be due to the differences in mutation rates, whereby the original authors typically used rates derived from ancient calibrations, while in this study we attempted to use more recent calibration dates to correct for the potential time-dependency of molecular rates (Ho et al., 2011).

Of course, we acknowledge that the signal of deviation from neutrality we detected may, in some cases, be the result of a selective sweep and not a rapid expansion. This signal could be distinguished by incorporating multi-locus data; nevertheless, given that a variety of species in this study showed similar genealogical patterns consistent with demographic expansion, it seems likely that most of them did indeed experience demographic changes associated with the end of the LGM, rather than selective sweeps. Moreover, distinctive haplotypes were found in several species networks (Pelvetia canaliculata, Pomatoschistus minutus, Owenia fusiformis and Pectinaria koreni) to the south of where the Eurasian ice sheet is proposed to have extended during the LGM (Fig. 1). This finding suggests populations of these species may have survived in southern glacial refugia; though, as pointed out by some of the original authors, deep sequence divergences in some species (e.g., O. fusiformis and P. korena) and the lack of a species-specific molecular clock calibration makes inferences about refugia challenging (Jolly et al., 2005; Jolly et al., 2006).

It is difficult to suggest an explanation for the two expansions estimated to have pre-dated the LGM (using mismatch analysis), but which fall within the last glacial period. This pattern of pre-LGM expansion has also been reported in a number of previous studies for a variety of marine taxa (e.g., Hoarau et al., 2007; Marko et al., 2010; Ni et al., 2014; Almada et al., 2017). One potential explanation for this pattern is that sea level during the last glacial cycle did not decrease uniformly towards the level observed at the LGM, but oscillated rapidly over a period of 60 Ka to 30 Ka (see Fig. 3A in Lambeck, Esat & Potter, 2002). Therefore, it may be possible that we are detecting the signature of a population expansion during one of these sudden increases in sea level during the last glacial period. Alternatively, as the BSP analysis inferred a post-LGM expansion for these two datasets, this could be a limitation associated with the mismatch analysis approach, which does not consider genealogy, and may, therefore, produce a less precise estimation. In addition, the sample of genetic diversity for this species may not be representative (Karl et al., 2012) or the genetic signal we detected may have been the result of a selective sweep and not a rapid expansion.

The use of single marker mtDNA genealogies and coalescence theory can introduce challenges associated with the interpretation of data and these limitations should be acknowledged (Karl et al., 2012). For example, the populations under study may have experienced multiple episodes of growth and decline; however, only the most recent expansion event can be detected using coalescence analysis and, in some cases, these events may not be sufficiently severe to be detected (Karl et al., 2012). In addition, coalescent histories can differ amongst loci because they can experience mutation and drift independently. Therefore, analysis of a single gene only gives insight into the coalescent history of that locus, which may not always be representative of population history. Analysis of multiple loci and genomics would help to alleviate these concerns, and would likely provide enhanced resolution for exploring the phylogeography of northeast Atlantic marine fauna.

Although population expansions were detected in a number of species in this study and also in the wider literature, populations of other marine species, including five from this study, have been found to remain stable throughout the LGM. As previously reported, not all coastal marine taxa appear prone to demographic changes during or after ice ages (Janko et al., 2007; Marko et al., 2010; Olsen et al., 2010). It is also important to acknowledge that earlier events in the Pleistocene and more ancient events that pre-date the Pleistocene may have helped shape the contemporary patterns of genealogical structure observed in this study.

Conclusion

The findings of this meta-analysis indicate that species in the northeast Atlantic do not show a uniform pattern of phylogeography, but rather a mixture of complex contemporary genealogical structure. Reanalysis of demographic histories indicated that a large proportion of the species included in this study have experienced post-LGM expansions, supporting the general expectation that rapid population expansions occurred after the LGM as the ice sheets started to retreat (Hewitt, 2000; Hewitt, 2004). This suggests that regional extirpation during the LGM appears to be a common biogeographic history for many northeast Atlantic marine taxa. However, improvements in mutation rate estimates, as well as the incorporation of multi-locus markers and genomics, would likely provide greater accuracy and resolution for overcoming the challenges associated with single mtDNA genealogies, and for improving our understanding of phylogeography in the northeast Atlantic Ocean.

Supplemental Information

Table S1 Mutation rates

Click here for additional data file.

Supplemental Information 1 Jenkins et al PRISMA checklist

Click here for additional data file.

Figure S1 PRISMA diagram

Click here for additional data file.

Figure S2 Haplotype networks (all species/lineages)

Click here for additional data file.

Figure S3 Mismatch graphs (all species/lineages)

Click here for additional data file.

Supplemental Information 2 Raw FASTA files for each dataset used in this study

Click here for additional data file.

We thank the many authors who kindly provided or contributed to the original datasets used in the meta-analysis. We also thank SYW Ho (Sydney), CN Roterman (Oxford) and KF Thompson (Exeter) for insightful comments on the manuscript, and GW Freeman (Exeter) for species illustrations.

Additional Information and Declarations

Competing Interests

Author Contributions

Data Availability

Rita Castilho is an Academic Editor for PeerJ.

Tom L. Jenkins conceived and designed the experiments, analyzed the data, prepared figures and/or tables, authored or reviewed drafts of the paper, approved the final draft.

Rita Castilho and Jamie R. Stevens conceived and designed the experiments, authored or reviewed drafts of the paper, approved the final draft.

The following information was supplied regarding data availability:

The raw data is available in the Supplemental File.

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
