# Peer review of "Meta-analysis of northeast Atlantic marine taxa shows contrasting phylogeographic patterns following post-LGM expansions"

_PeerJ, doi:10.7717/peerj.5684_

## Round 0.1 · original submission · Major Revisions

All the reviewers are positive about your paper and make constructive suggestions. However, two recommend major revision. Please consider their suggestions positively and as comprehensively as possible and re-submit as soon as you can.

Reviewer 1 ·

Basic reporting

no comment

Experimental design

no comment

Validity of the findings

no comment

Additional comments

This is on the whole a solid meta-analytical study of mtDNA marine population genetics that is aware of its limitations, but works appropriately within those bounds. It is well written and presented and will be of utility to anyone hoping to gain insights into patterns of connectivity and recent demography in populations of NE Atlantic marine fauna. Below are some areas that could be improved however:

1) The formula used to estimate the timing of population expansion assumes an exponential expansion and produces a single date for the expansion. Is there a reason why you haven't considered using bayesian skyline plots (e.g. in Beast) which provides an age bounded within a 95% CI to complement your estimates of demographic expansion? This might offer more nuanced results.

2) In the methods, you haven't made clear with which software the Tau parameter was calculated.

3) On line 90 you state that the "British-Irish ice sheet led to a drastic drop in sea levels in the English channel". Are you saying that the lower sea level in the English channel is a local effect? One would presume that lower sea levels at the LGM were a global response to the glaciation.

4) On lines 233, 234, 235 you appear to differentiate between unimodal and skewed unimodal mismatch distributions with the former associated with sudden population expansions and the latter associated with recent expansions or bottlenecks. I'm not aware of this - can you provide a reference to the relevant research that demonstrates this? My understanding of mismatch distributions is that at best they can indicate either recent expansions/bottlenecks or the lack thereof - but not how sudden this expansion was.

5) On lines 240-248 you appear to suggest that bimodal mismatches indicate two or more distinctive lineages. This may be true, but those lineages do not necessarily mean that you're dealing with separate cryptic species or "inter-clade" differences necessitating dividing up your datasets in the demographic analyses. As mitochondrial DNA can be considered a single locus with effectively no recombination, divergent lineages can still be produced in large, panmictic populations - as I think you state later on lines 291-293. The only real indication that the divergent lineages may be indicative of cryptic diversity, thus warranting separate demographic analyses is where these lineages are geographically separated and not sympatric. Nevertheless it doesn't hurt to run the analyses with the separate lineages where there is a reasonable expectation that they may have separate demographic histories.

6) Lines 262-263 belongs in the discussion - i.e. "suggestive of population expansions as the ice started to retreat from its maximum positions"

7) Lines 269-270: The sentence - "Cases of species-specific unique haplotypes were evident at certain locations across the sampling range" could be written more clearly. Are you saying that in some species, certain regions exhibited unique haplotypes?

8) Lines 312-313: Is there circular reasoning here: "given that many species/lineages in this study showed a pattern consistent with demographic expansion, it seems likely that a large proportion of them did indeed experience a rapid population expansion...". I think you mean that a generally similar pattern in mtDNA diversity across a variety of species points toward demographic changes associated with the end of the LGM, rather than elective sweeps. That being said - dramatic climate changes could also be a trigger for selective sweeps - which would also be interesting...

9) Line 317: I think "have" should be inserted between "may" and "survived".

10) Line 330: There are also other things that could explain the result, such as your sample of genetic diversity for this species/lineage not being representative (see Karl et al. 2012) - or that mutation rate estimates are wrong or too conservative for this species, for example.

Reviewer 2 ·

Basic reporting

In general, I found this manuscript clear and concise with a well-explain aim to achieve, and interesting conclusions. In addition, the manuscript is clearly written in professional, unambiguous language throughout. It is clear, coherent and used a professionally presented format. The manuscript is has a very good structure and show context of the current knowledge and question of interest. However, I do have some suggestions for the authors, which they may wish to consider including and that would strengthen the paper. If there is a weakness, it is in the statistical analysis (as I have noted below) which should be improved upon before acceptance.

Tables and figure are relevant and well explained. In addition, I found the Figure 4 is really informative and well-describe the main conclusion of this manuscript. The inclusion of some examples of the different haplotype networks shapes and mismatch distributions are very useful. I suggest use the same species name order in both tables (Table 1 and 2) which will make easier the comparing between both.

Line 93-95 : It would make easier for the reader to make a reference to the map and include the names in the map (Hurd Deep/Iberia) to clarify which refugia are pointed out.

Your introduction covers the state to date knowledge of comparative phylogeography but it may need more detail about the importance of this field. For example in Para 1: It would be good to have a sentence about the importance of using comparative phylogeography also in terms of management and conservation purposes. (Toonen et al. 2011; Villamor et al. 2014; Liggins et al. 2016; Pelc et al. 2009; Von der Heyden et al. 2014).

Line 96-100 : Reference to this part, there are many approaches for multispecies analysis. It would be interesting to add some references to support this point.
I thank you for providing specifically details on how to access to the raw data and the inclusion/exclusion criteria that you used. I have a few questions that may be worth to also explain in the methodology section.

References:

Liggins L, Treml EA, Possingham HP, Riginos C (2016) Seascape features, rather than dispersal traits, predict spatial genetic patterns in co‐distributed reef fishes. Journal of biogeography 43, 256-267.
Pelc RA, Warner RR, Gaines SD (2009) Geographical patterns of genetic structure in marine species with contrasting life histories. Journal of Biogeography 36, 1881-1890.
Toonen RJ, Andrews KR, Baums IB, et al. (2011) Defining Boundaries for Ecosystem-Based Management: A Multispecies Case Study of Marine Connectivity across the Hawaiian Archipelago. Journal of Marine Biology 2011.
Villamor A, Costantini F, Abbiati M (2014) Genetic Structuring across Marine Biogeographic Boundaries in Rocky Shore Invertebrates. PLOS ONE 9, e101135.
Von der Heyden S, Beger M, Toonen RJ, et al. (2014) The application of genetics to marine management and conservation: examples from the Indo-Pacific. Bulletin of Marine Science 90, 123-158.

Experimental design

The research question and main goals are defined and well-introduce. However, I include below some suggestions about where it may be worth to improve the introduction of your research question.

Line 96-100: Reference to this part, there are many approaches for multispecies analysis. It would be interesting to add some references to support this point.

Line 103-105: This point is clear in the first paragraph of the introduction. In this paragraph, it would be more interesting to see some hypothesis or expectations based on previous findings in the Atlantic ocean (Marko et al. 2010) or other similar studies around the world, to emphasize the importance of your question.

Line 105-106: It seem that there are previous studies on the western part of the Atlantic ocean but nothing in the Northeast Atlantic. It would be good to expand upon the knowledge gap being filled.

One of the strengths of this study is the wide literature searched and data reanalysis performed which are really well described and able to replicate. The literature search and the inclusion of the final studies is a considerable piece of work. I have a few minor comments about the species information included in the analysis that it may need to be clarify (from line 126 to 138):
- Are they all covering the whole distribution range of the species? If you only include the sampling locations in the area of interest (northeaster Atlantic) you may need to clarify.
- Have you include only endemic species?

Validity of the findings

The rationale and the benefit of using the literature for this meta-analysis are meaningful and well explained. It is clearly beneficial the used the existing genetic dataset to study the main question of this manuscript. I commend the authors for their extensive literature review and work in reanalysing the genetic dataset with a standardised methodology to uniform the wide range of methods used in previous studies.

On the other side, I have a major comment about the statistical methodology used. In this study, a mismatch distribution approach have been used and it seem to be useful in order to estimate the time of expansion in case it was detected. However, in order to examine the historical demography, there are many other studies supporting the used of Bayesian inference to improve the precision of the results (e.g Crandall et al. 2010; Villamor et al 2014). It has been found that the mismatch analysis provide only analytical approximation of substitution patterns expected from a spatial expansion, without considering genealogy, therefore it may yield less precise estimation. I would recommend considering the use of Bayesian Skyline Plots (BSPs) using the software BEAST v. 1.7.5 (Drummond and Rambaut (2007).

I thank you for providing the mutations rates references in the supplementary material, however the divergence dates used for the Carcinus maernas for example, were based on Crandall et al. 2012. And there are considerably higher than the ones reported by Roman and Palumbi 2004 (1.4-2.3% per Mya). Please can you develop why you used this mutation rates instead.

Discussion is well stated, linked to original research questions and limited to the supporting results. I have some minor comments about your first (Line 111) aim but I want to highlight that the other two aims are neatly discussed and linked to the original questions. I really appreciate the discussion about the possible limitations and future directions.

One of the aims in the introduction was (1) to identify commonalties (or otherwise) in contemporary genetic structure and you discuss some finding in the first paragraph of the discussion. I suggest that you clarify with a starting sentence in the paragraph (Line 268) if you did find some commonalities or not. In addition, in your report, you included interesting information about the pelagic larval duration for the studied species (Table 1). However, this information is not incorporated in any part of your manuscript and it would be worth developing some ideas. It would be interesting to mention if you observed the level of population structure (Fst/Djost) or/and the haplotype network found were related to the pelagic larval duration or otherwise, there was no relation. Speculation is welcome, but should be identified as such. Several studies have shown that PLD is a weak predictor of the genetic differentiation (Weersing & Toonen 2009;Kelly & Palumbi 2010; Riginos et al. 2011).

References:
Kelly RP, Palumbi SR (2010) Genetic structure among 50 species of the northeastern Pacific rocky intertidal community. PLOS ONE 5, e8594.
Riginos C, Douglas KE, Jin Y, Shanahan DF, Treml EA (2011) Effects of geography and life history traits on genetic differentiation in benthic marine fishes. Ecography 34, 566-575.
Weersing K, Toonen RJ (2009) Population genetics, larval dispersal, and connectivity in marine systems. Marine Ecology Progress Series 393, 1-12.

Annotated reviews are not available for download in order to protect the identity of reviewers who chose to remain anonymous.

Reviewer 3 ·

Basic reporting

Well-presented and clearly written, with appropriate background / context and use of the literature, although a short summary of possible refugial areas for NE Atlantic species would have made the Introduction more complete.

Experimental design

Within scope of the journal. Good meta-analysis of NE Atlantic phylogeographic studies on animals. Approach clearly described.

Validity of the findings

Findings valid (if mostly unsurprising), and handled well in the Discussion. Could possibly have included a comparison with findings from non-animal taxa (mostly seaweeds), which may present different patterns.

---

## Round 0.2 · Minor Revisions

Thank you for the careful revision of this paper. One referee makes a suggestion regarding a figure legend. Perhaps you can attend to that and do another careful proof read and then submit a final version soon?

Reviewer 1 ·

Basic reporting

I have little to write here. The authors have been thorough, clear and concise. The paper is well structured, the analyses appropriate and the figures are good.

Experimental design

The study fits within the scope of the publication, with a relevant and timely assessment of past climate changes on the genetic demographic structure of a variety of marine fauna in the NE Atlantic. The authors have taken onboard suggestions by the reviewers to use a bayesian approach to estimating demographic history.

Validity of the findings

The authors conclusions are appropriate and reasonable given their findings. They provide useful new evidence filling a regional gap in our understanding of how past climate changes have influenced marine populations.

Additional comments

I have little to comment here, given that my queries and comments appear to have been adequately addressed. This is a good paper. The only thing I would note is that the figure caption for Fig. 5 (the BSP plots) does not clarify what the y axis is, nor what the dotted and solid lines are. Thus I recommend accept with this one minor revision.

Reviewer 2 ·

Basic reporting

No comment

Experimental design

No comment

Validity of the findings

No comment

Additional comments

This meta-analysis of marine taxa in NE Atlantic is a very interesting piece of work with interesting results that help to understand patterns of connectivity and the influence of historical events in the spatial genetic pattern of the marine fauna. Well-written and presented and good definition of the limitations of the study.

One of my major concerns in the previous revision was the formula used for the time of expansion. The inclusion of bayesian skyline plots gives an important statistical support to the methodology.

I am more than satisfied by the authors' responses and I recommend this manuscript to be accepted. I appreciate the inclusion of the suggestions and the clarity in the responses and congratulated the authors for the work undertook.

---

## Round 0.3 · accepted · Accept

Nice work, and thank you for addressing the referees detailed and thorough comments well.

In proofs, please consider

(1) edit lines 83 and 84 to say "In the northeast Atlantic, the ice sheets .. as Britain and Ireland" (omit 'it is thought that', clarify both main islands were largely covered).
(2) replace uses of / with and, or, delete one of the words, or whatever is meant (e.g. line 326, 380).
(3) try to get Tables portrait format.

As all bar one species are subtidal, presumably there were other potential refugia offshore (Rockall, Faeroes). The present model seems only to consider coastal.

I spawned Pomatoschistus minutus in Scotland in aquaria, the planktonic larvae stage extended 1-2 weeks. As PLD are temperature dependent then it may vary a lot within a species range. Ideally I'd include amibent temperature for each study and report as degree-days.

Thank you for submitting your work to PeerJ.

#